# Continent-wide tree fecundity driven by indirect climate effects

James S. Clark [1,2✉], Robert Andrus[3], Melaine Aubry-Kientz[4], Yves Bergeron [5], Michal Bogdziewicz [6], Don C. Bragg[7], Dale Brockway[8], Natalie L. Cleavitt[9], Susan Cohen[10], Benoit Courbaud[2], Robert Daley[11], Adrian J. Das [12], Michael Dietze[13], Timothy J. Fahey[8], Istem Fer[14], Jerry F. Franklin[15], Catherine A. Gehring[16], Gregory S. Gilbert [17], Cathryn H. Greenberg[18], Qinfeng Guo [19], Janneke HilleRisLambers[20], Ines Ibanez [21], Jill Johnstone[22], Christopher L. Kilner [1], Johannes Knops[23], Walter D. Koenig [24], Georges Kunstler[2], Jalene M. LaMontagne [25], Kristin L. Legg[11], Jordan Luongo[1], James A. Lutz [26], Diana Macias[27], Eliot J. B. McIntire[28], Yassine Messaoud[29], Christopher M. Moore[30], Emily Moran[3], Jonathan A. Myers [31], Orrin B. Myers[32], Chase Nunez [33], Robert Parmenter [34], Sam Pearse[35], Scott Pearson [36], Renata Poulton-Kamakura[1], Ethan Ready[1], Miranda D. Redmond[37], Chantal D. Reid[1], Kyle C. Rodman [2], C. Lane Scher[1], William H. Schlesinger[1], Amanda M. Schwantes[38], Erin Shanahan[11], Shubhi Sharma[1], Michael A. Steele[39], Nathan L. Stephenson [12], Samantha Sutton[1], Jennifer J. Swenson[1], Margaret Swift[1], Thomas T. Veblen [2], Amy V. Whipple[16], Thomas G. Whitham[16], Andreas P. Wion[37], Kai Zhu [17] & Roman Zlotin[40]

Indirect climate effects on tree fecundity that come through variation in size and growth (climate-condition interactions) are not currently part of models used to predict future forests. Trends in species abundances predicted from meta-analyses and species distribution models will be misleading if they depend on the conditions of individuals. Here we find from a synthesis of tree species in North America that climate-condition interactions dominate responses through two pathways, i) effects of growth that depend on climate, and ii) effects of climate that depend on tree size. Because tree fecundity first increases and then declines with size, climate change that stimulates growth promotes a shift of small trees to more fecund sizes, but the opposite can be true for large sizes. Change the depresses growth also affects fecundity. We find a biogeographic divide, with these interactions reducing fecundity in the West and increasing it in the East. Continental-scale responses of these forests are thus driven largely by indirect effects, recommending management for climate change that considers multiple demographic rates.

A list of author affiliations appears at the end of the paper.

The composition and structure of twenty-first century forests will depend on the seed production needed for tree populations to keep pace with climate change. North America is warming and drying out in much of the West. The dramatic impacts include large-scale die-backs[1–3] that are transforming size-species structure[4,5]. But the decade-scale trends will depend on the regeneration that follows tree death. Fecundity will determine the capacity of trees to disperse seed to the shifting habitats where they can survive in the future[6–8]; risks to each species depend not only on the current distribution of fecundity but also on its trajectory[4,9–13]. As with many ecological processes[14–16], noisy, spatially variable fecundity trends are hard to quantify[8,17], but this is only the first problem. Attributing trends to environmental variables is complicated by individual size, growth, and resource access[18–20]. Conservation efforts must anticipate not just the direct climate effects on this trajectory but also the indirect effects as growth and changing size structure also affect fecundity. Because it has thus far been impossible to estimate at continental scales, fecundity is the only major demographic process that lacks field-based estimates in models of vegetation change[5,21,22]. To address these challenges, we built the continental Masting Inference and Forecasting (MASTIF) network of primary data (Fig. 1), and we developed trend attribution (TA) to quantify climate impacts, as modulated by the condition of the organisms themselves. Application to the MASTIF network shows that indirect effects dominate, operating through stand structure and growth.

Although a substantial climate-impacts literature has focused on growth responses to short-term (interannual to a decade) climate fluctuations[3,23], this focus is not based on evidence that fecundity effects are of secondary importance. The emphasis on tree growth comes in part from the facts that (i) data are widely available from inventory plots and tree-ring records, (ii) where absent they can often be obtained from tree rings for long periods in the past, and (iii) growth anomalies can often be at least partly explained by climate anomalies[3,23]. By contrast, fecundity is not directly observed for most species and habitats, data accumulate slowly and with substantial investment, and the effects of climate anomalies can be overwhelmed by nonlinear, internal feedbacks on reproductive effort[18,22,24–27]. Although critical for population dynamics, a limited role for fecundity could be interpreted from stand simulators that stabilize dynamics by assuming an external seed pool[28–30]. This is done both for lack of estimates, but also because the contribution of fecundity to dynamics is too poorly understood to construct models that allow species to coexist; even models used to explore effects on species diversity depend on the assumption that seeds remain available even when adults are not[30]. Foundational understanding of population growth makes clear that fecundity contributes directly to fitness[31–33], while the change in size can do so only indirectly. While evidence points to the direct importance of fecundity for future forests, we show here that it must be coupled with the indirect role of tree growth.

We identify and quantify the effects of climate change on fecundity at the continental scale, including the climate-condition interaction (CCI) that require individual-scale observations (Fig. 2). We hypothesized that climate change will be experienced by organisms, each in its own way. We use the term CCIs to include, for example, moisture effects that differ for deep-rooted adults and small saplings[26,34] and temperature effects that depend on light availability[35]. If CCIs are important, then they must be quantified at the scale of individuals. We find biogeographic differences in the indirect effects of climate change, slowing fecundity change in the West and increasing it in the East.

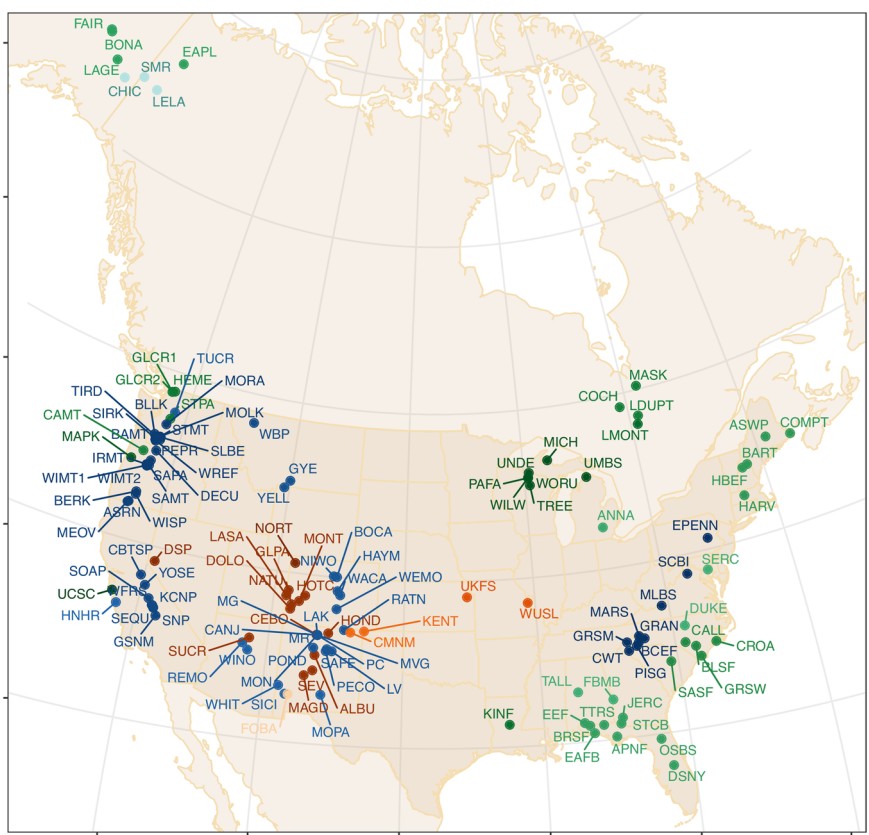

**Fig. 1 Longitudinal sites in the MASTIF network.** Colors match ecoregions in Fig. 3. Sites are listed by ecoregion in the Supplementary Data 1.

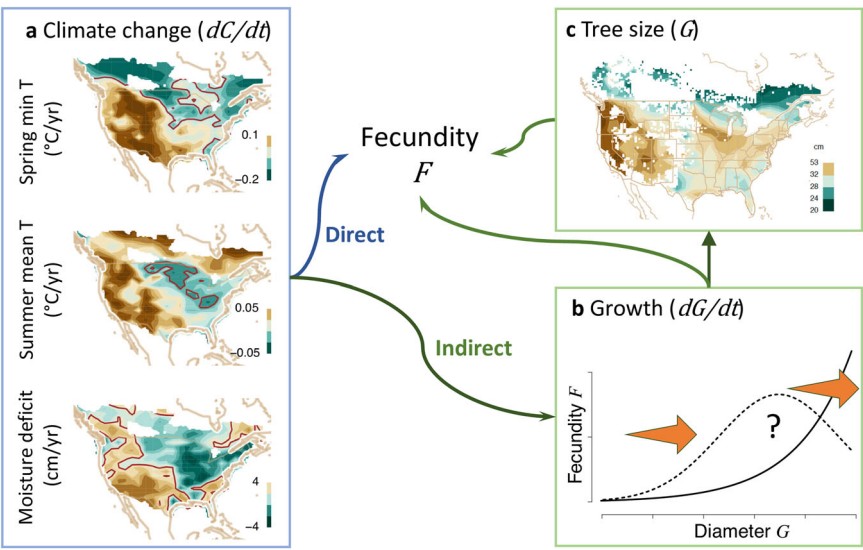

**Fig. 2 Trend attribution (TA) includes direct and indirect pathways for terms in Eq. (1). a** Trends in climate variables (since 1990) include minimum $T$ in spring, mean summer $T$, and moisture deficit ($D$ = cumulative monthly PET-P). The brown contour separates positive and negative trends. Shaded contours are green (decreasing) to brown (increasing). **b** Indirect effects have two elements. An arrow from **b** to $F$ includes a growth effect (d$G$/d$t$) and a climate-growth interaction ($C$ × d$G$/d$t$). An arrow from **b** to **c** depends on the uncertain relationship between tree diameter $G$ and fecundity $F$ shown in panel (**b**). If fecundity continues to increase with tree size (solid line in **b**), then accelerated growth (orange arrows are d$G$/d$t$) moves trees into more productive size classes, but not if fecundity eventually declines (dashed line). **c** Average diameter of trees (restricted to trees >20 cm) is high in the West, meaning that the effects of tree growth depend on [if] fecundity continues to increase or declines with size in panel (**b**). The effect of size on fecundity (arrow from **c** to $F$) comes through an interaction with climate ($G$ × d$C$/d$t$ in Eq. (1)).

## Results

TA was developed to quantify change that emerges from both direct and indirect effects and that are not amenable to traditional time-series methods. Effects of climate change are increasingly apparent, including shifts in phenology[36] and species range limits[37,38]. By contrast, the time series of species-abundance data typically lack a clear signal[16,39,40]. No ecological process suffers more from the signal-to-noise problem than seed production, where quasiperiodic, order-of-magnitude variation from year to year and tree to tree[8,18,19,25,41,42] can bury long-term trends. Autoregression models assume a fixed periodicity, but mast intervals are not fixed, not even within an individual[25,35,43]. There are as many time series as there are trees (>10$^5$ in this case), but they must be modeled together because there is dependence. Data are non-Gaussian (including zeros for immature trees and failed crops). Trends estimated by meta-analysis may not be comparable across studies due to divergent methods and transformations intended to force non-Gaussian data into traditional time-series models[20]. Efforts to determine whether a species is increasing or decreasing are further challenged by the uneven distribution of publications. A standard trend analysis of our sites (Fig. 3a) shows not one trend but rather a broad range, with most species (bars in Fig. 3b) increasing in some habitats while decreasing in others. Estimates are readily biased[16] due to haphazard habitat coverage (Fig. 3a).

The MASTIF network includes the primary tree-year data (a given tree in a given year) that are needed to estimate change and the contributions of CCI. Data include the canopy environment (fully exposed to deep shade) and tree size, recognizing that accelerated growth can speed reproductive maturity[44]. Ecological studies have assumed that fecundity continues to increase with stem diameter (Fig. 2b, solid line)[45–49]. However, horticultural practice suggests declines in large trees (Fig. 2b, dashed line), but the literature is limited[50]. Several ecological studies also suggest

declines[20,51–53], but inference suffers from few observations of large trees. The MASTIF network offers a broad range of sizes combined here with weighted regression methods that allowed us to quantify the effects of both maturation and eventual fecundity declines (see "Methods" section). Fecundity data include seed traps (STs) and crop counts (CCs) from longitudinal studies (Fig. 3a) and opportunistically through the iNaturalist MASTIF [https://www.inaturalist.org/projects/mastif] project, including 2,566,594 tree-years from 123 species. The dynamic model accommodates non-Gaussian data and serial and intertree dependence, with full uncertainty for data, model miss-specification, and parameters[20]. Continental prediction used 7,723,671 trees from inventory plots (see "Methods" section).

TA was developed to decompose direct and indirect effects on change. For transparency, three climate variables in Fig. 2a are represented here by a single state variable $C$. To evaluate community-wide effects, we report on log (proportionate) fecundity change,

$$\frac{df}{dt} = \frac{1}{F}\frac{dF}{dt} = \frac{d\log F}{dt}$$

where $F$ is seeds per tree per year. Proportionate change d$f$/d$t$ is analyzed because we are interested in effects on species of both high and low fecundity. Analysis of absolute change d$F$/d$t$ would be dominated by the few species that produce the most seeds. To obtain community change, we average these proportionate changes over all trees on a plot.

TA entails (i) model fitting and (ii) trend decomposition. Model fitting estimates responses as fitted coefficients. These responses are main effects of climate, $\partial f/\partial C$, and size (diameter $G$), $\partial f/\partial G$, and their interaction $\partial f/\partial(GC)$. Trend decomposition combines these responses with dense information on the environment, including individual states ($G$, $C$) and their rates of

**a Sites**

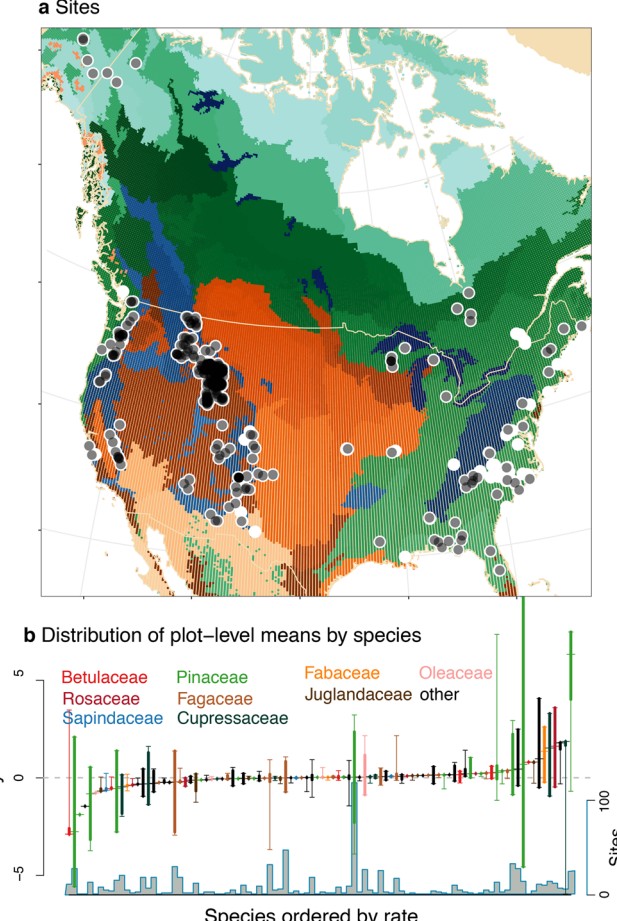

**b Distribution of plot–level means by species**

**Fig. 3 Sites and species trends. a** Longitudinal studies in black (opacity proportional to numbers of sites) and opportunistic in white. Shaded ecoregions are desert/shrub/grass (browns), montane (blues), and mixed forest (greens). **b** Trends in mean log (proportionate) fecundity by species from sites in **a** range from negative (declining) to positive. As would be the case for any meta-analysis, the time scales for which trends are evaluated vary (see "Methods" section). Species belong to color-coded families below that are listed in Supplementary Data 2. There is no relationship with phylogeny (i.e., no trend in box color from left to right). Summaries in **b** include mean (crosshairs), 95% of site means (bold line), and range of site means (whiskers). The number of sites (*n*) contributing to **b** is shown below.

change (d$G$/d$t$, d$C$/d$t$) (Fig. 2), summarized with three terms,

$$\frac{df}{dt} = \overbrace{\frac{\partial f}{\partial C}\frac{dC}{dt}}^{\text{direct}} + \overbrace{\underbrace{\frac{\partial f}{\partial G}\frac{dG}{dt}}_{\text{growth effect}} + \underbrace{\frac{\partial f}{\partial (GC)}\left(G\frac{dC}{dt} + C\frac{dG}{dt}\right)}_{\text{growth-climate interactions}}}^{\text{indirect}} + \underbrace{\gamma}_{\substack{\text{neither growth}\\\text{nor climate}}}$$

(1)

The first two terms are both main effects that depend respectively on rates of change in climate (d$C$/d$t$) and tree growth (d$G$/d$t$). The direct effect (first term in Eq. (1)) combines the climate response with the rate of climate change (Fig. 2a). This direct effect is followed by three terms that contribute the indirect effects of size and growth. The third term holds their interaction ($GC$) as products of rates (d$G$/d$t$, d$C$/d$t$) and states ($G$, $C$). [Again, $C$ is a placeholder for multiple environmental variables (see "Methods" section)]. These are (i) the size-dependent effects of climate change and (ii) the climate-dependent effects of growth.

The residual $\gamma$ allows effects that are not attributed to other terms. The full effect of a climate variable $C$ combines the direct term 1 with its indirect effects in the second and third terms.

Indirect terms are CCI, incorporating climate effects that are modulated by tree size, $G \times dC/dt$, as when large, deep-rooted trees experience drought differently from saplings. Conversely, a change in growth rate has effects that can vary with climate, also a CCI, $C \times dG/dt$. The indirect effects through growth $g(C) = dG/dt$ are not shown in Eq. (1), but are given in the "Methods" section. Depending on how fecundity changes with size (Fig. 2b), climate stimulation of growth can move small trees into more fecund size classes. If fecundity eventually declines with size, growth has the opposite effect on large trees.

TA in Eq. (1) starts from a notion similar to "climate velocity"[9,54], which replaces fecundity in Eq. (1) with distance $x$ as d$x$/d$t$ = d$x$/d$C$ × d$C$/d$t$. Rather than distance-over-time in climate velocity, TA decomposes the climate and size contributions to fecundity trends over time. It relies on extracting the smooth trends from volatile seed production data. The terms in Eq. (1) are available each as a predictive distribution for each tree. There is an average over trees for each inventory plot. The climate effects on fecundity trends differ from sensitivity to interannual variation[8,17–19,25,35]. A species that reduces seed production in dry years (negative response to moisture deficit $D_{j,t}$ in Table S2.1) may not suffer from dry climates in general—indeed, the capacity to reallocate under fluctuating conditions can be adaptive. A negative effect in TA means that species and size classes of the current forests produce less seed under the decade-scale trends occurring now, based on responses across climate and habitat variation.

**Indirect effects dominate response**. TA shows that continent-wide trends are dominated by indirect effects. Maps of these effect in Fig. 4 have different scales, which is necessary to show the geographic patterns within maps; the scale differences must be considered when comparing maps. The direct responses in Fig. 4a are transformed by the heterogeneity of climate trends (Fig. 4b) and then, indirectly, through tree growth (Fig. 4c). The responses in Fig. 4a are positive where trees dominate that have high mean fecundity responses. For example, trees that are most fecund under high spring $T$ (Fig. 4a, top) and moisture deficit $D$ (bottom) are concentrated in the Northwest (NW) and Southeast (SE).

The direct responses in Fig. 4a are multiplied by heterogeneous climate change (Fig. 2a) to yield the direct effects (Fig. 4b). It is important to recognize that a positive effect of climate change occurs wherever the response to climate and the direction of climate change have the same sign. For example, the negative direct effects of spring $T$ in the Northeast (NE) and Southwest (SW) (Fig. 4b, top) result from opposing forces: in the NE, mostly positive responses (Fig. 4a, top) combine with a negative spring $T$ trend (Fig. 2a), that is, (+) × (−) = (−). In the SW, negative responses combine with a positive spring $T$ trend, that is, (−) × (+) = (−). Between is a swath of positive effects stretching from the NW toward the SE (Fig. 4b, top), where positive responses overlap with rising spring $T$ (Fig. 2a). The direct effects of other climate variables are near zero or negative for summer $T$ (Fig. 4b, middle). The limited direct effects of moisture deficit $D$ is apparent from the scale differences for maps in Fig. 4b.

The foregoing direct effects are overwhelmed by the indirect effects (contrast scales in Fig. 4b, c), both as main effects on growth in the second term and interactions in the third term of Eq. (1). Whereas the full effects contribute to a positive east/ negative west divide in the effects of spring $T$ (Fig. 4c, top) and moisture deficit $D$ (Fig. 4c, bottom), the contribution of summer

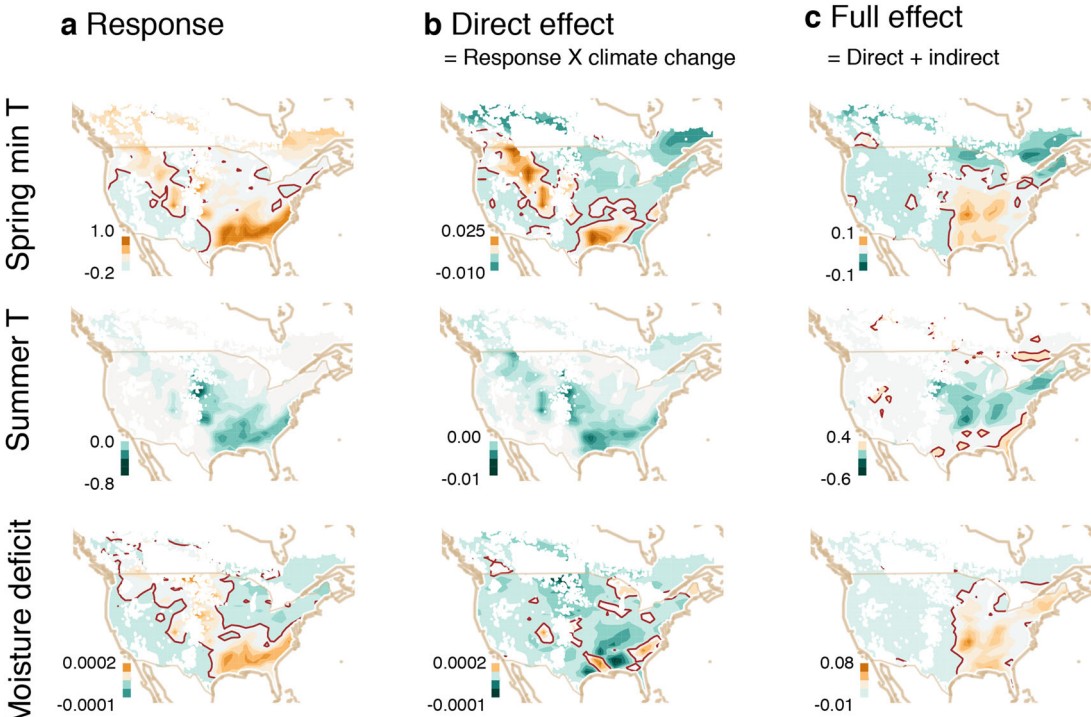

**Fig. 4 Continent-wide causes for fecundity trends.** Shaded contours are green (decreasing) to brown (increasing). Responses to climate variables ($\partial f/\partial C$) in **a** are multiplied by climate change ($\times dC/dt$) in Fig. 2a to give the direct effect ($\partial f/\partial C \times dC/dt$) in panel (**b**) (first term of Eq. (1)). The direct effect in **b** is added to the indirect effect that comes through tree growth (terms 2 and 3 of Eq. (1)) to give the full effect in panel (**c**). Units are proportionate change in fecundity per °C or per mm[-month] moisture deficit in panel (**a**) and per year in panels (**b**, **c**). White areas lack inventory plots. The brown contour indicates zero.

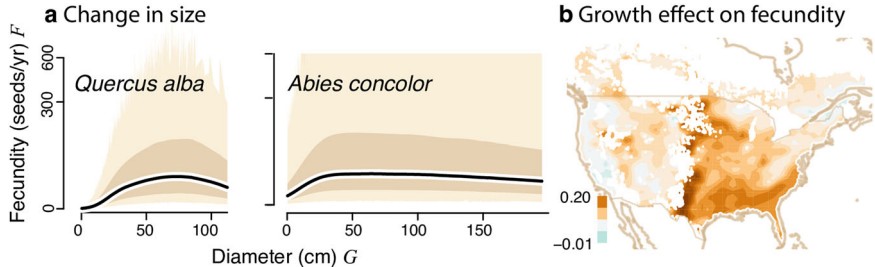

**Fig. 5 Indirect effects produce an East–West contrast. a** Fecundity rising then falling with size, for a common eastern hardwood (*Q. alba*) and western conifer (*A. concolor*) plotted on the square root scale. The predictive mean (black line) is bounded by the 90% credible interval (dark shading) and the 90% predictive interval (light shading) over all tree-years. **b** The growth effect includes terms in Eq. (1) that are multiplied by $dG/dt$, that is, $\partial f/\partial G + \partial f/\partial (GC) \times C$ (units are log $f$/yr). Averages shift from positive in the East to near zero or negative in the West, where more trees are near or past the diameter where growth stimulation increases fecundity.

$T$ is primarily negative in the East (Fig. 4c, center). Positive effects in the East come predominantly through spring $T$ (Fig. 4c, top), which is transparent because both responses and climate trends tend to share the same sign (both positive: Figs. 2a and 4a, top). The full effects could not have been anticipated from the direct responses because they require consideration of how growth responds to climate and the effect of size and growth on fecundity.

To understand continental responses and the large differences between maps in Fig. 4b, c requires the decomposition of effects, which is available through the individual terms in Eq. (1). An important contributor to these differences is the pervasive fecundity declines that our analysis found for large trees (Fig. 5a). Due to management and species traits, growth stimulation in the

East speeds the transition of small trees to larger, more fecund size classes (Fig. 5b). Conversely, much of the West supports trees that have passed this size. The East–West differences are amplified by maturation, which is increasing the probability of seed production in the East, but not the West (Supplementary Fig. S2.3b). Declining fecundity in large trees (which are also older in the West, Supplementary Fig. S2.3a) does not necessarily come from physiological decline ("senescence") because [declines can result from] crown architectural changes also occur.

**Discussion**. Continent-wide impacts of climate change are being governed by the indirect effects that come through the condition of individuals. Global change science has steadily improved understanding of direct responses, including photosynthetic rates,

water use, and demography[25,55,56] and the abundances of species[40]. Lagging behind is the understanding of interactions and indirect effects[18]; individual responses to climate do not predict responses of canopies[57], just as species responses do not predict outcomes of competition[40]. However, climate variation operates on individual organisms that see that change differently. From the multitude of individual traits that affect climate response, we show that size and growth differences provide important insights on continent-wide change. By combining detailed habitat change with the individual condition (Fig. 4), TA shows the dominance of indirect effects.

The geographically coherent picture of change that emerges from TA contrasts with inconclusive results offered by meta-analysis of trends. For example, the conflicting interpretations from recent studies of insect abundances[16,58] reflect over-sensitivity to precisely which sites and species were included in each meta-analysis[58]. Indeed, simple trend analysis of our sites (Fig. 3b) is no more informative than that of ref. [58] (their Fig. 2), both showing that nearly every species is increasing and decreasing somewhere. Only long time series can provide reliable estimates of trends for erratic data, but the duration is not enough —the unrepresentative geographic distribution of sites precludes an interpretation of overall trends. TA does not attempt to extract signal or extrapolate from noisy data, [instead of] exploiting instead relationships between varying climate and individual condition. It benefits from dense geographic coverage of sites, but can provide insight without it, relying primarily on adequate coverage in covariate space rather than geographic space.

TA adds value to existing efforts because climate change is heterogeneous not only in rate but also in sign (Fig. 2). By exposing the trends masked by interannual and intertree volatility of seed production while incorporating effects of additional variables TA provides much-needed perspectives on patterns and processes that affect individuals. Because climate variables interact with one another and with the individual condition (CCI), models need to not only find coefficients for their effects (Fig. 4a) but also combine them with the changes in climate that are happening now (Fig. 4c). Current understanding suggests that fecundity is also responding to variables that could not be incorporated into this analysis, including changing $CO_2$[44], irradiance and clouds[59], and soils, depending only on data availability and distribution across sites.

TA can complement efforts based on stand simulators and species distribution models by quantifying contemporary change and extracting the reasons for it. For instance, because stand simulators rely on immigration to achieve species coexistence, fecundity estimates are mostly absent and without direct or indirect effects of climate that are based on field data. TA combines growth with fecundity estimates at the tree-year scale for understanding biogeographic consequences, thus offering an alternative perspective to stand simulators and map-based predictions of future biodiversity[60].

Climate change is driving fecundity in two directions across North America, predominantly [declining] negative in the West and [increasing] positive in the East. Rising temperatures and moisture deficits are negative contributions in the West [contributing to western declines], while seasonal temperature differences have opposing [effects] contributions in the East (Fig. 4c). The full effect differs from direct effects (Fig. 4b, c) due to indirect effects of climate on tree growth. Growth changes have limited impact on fecundity trends in the West because few trees are nearing maturity (Supplementary Fig. S2.2b), and fecundity has plateaued or is decreasing (Fig. 2c). By contrast, climate changes are accelerating change toward fecund size classes in the East.

The finding that fecundity can decline in large trees, with biogeographic consequences, does not diminish their contribution to biological diversity through microhabitats for wildlife[61]. Selective removals that promote uneven-aged structures can preserve microhabitats and promote canopy heterogeneity and light penetration that stimulates growth and fecundity[35]. Growth is not currently making a strong contribution to average trends in the West; however, management priorities can be guided by disaggregating these mean trends to understand their distribution across species at risk and/or valued for their ecosystem services.

The determination that indirect effects through individual condition can dominate biogeographic responses has immediate application in forestry and conservation. As an example, scientists and managers increasingly recognize that the challenges posed by continuing trends in climate cannot be addressed with traditional nursery practice or silvicultural treatments[62]. Managing for long-term trends (as opposed to the volatile interannual variation) must consider both the direct effect of climate-induced changes on growth and the indirect effects of these changes on fecundity. This knowledge is critical because size-species structure is often under the direct control of forest managers and conservation planners, especially in eastern North America[63], whereas climate is not[4]. TA offers concrete estimates of how fast these changes are happening now and which variables are responsible. While climate is not controllable by managers, the opportunity to influence indirect effects through stand structure can foster stronger connections between conservation planning and global change science.

## Methods

**Elements of TA.** Identifying biogeographic trends within volatile data required several innovations in the MASTIF model[20], building from multivariate state-space methods in previous applications[41,52]. Standard modeling options, such as generalized linear models and their derivatives, do not accommodate key features of the masting processes. First, multiple data types are not independent. Maturation status is binary with detection error, CCs are non-negative integers, also with detection error, and STs require a transport model (dispersal) linking traps to trees, and identification error in seed identification. Of course, a tree observed to bear seed, now or in the past, is known to be mature now. However, failure to observe seed does not mean that an individual is immature because there are detection errors and failed crop years[41,64].

Second, seed production is quasiperiodic within an individual (serial dependence), quasi-synchronous between individuals ("mast years"), [and] there is dependence on environmental variation, and massive variation within and between trees[41,53,65]. Autoregressive error structures (AR($p$) for $p$ lag terms) impose a rigid assumption of dependence that is not consistent with quasiperiodic variation that can drift between dominant cycles within the same individual over time[43]. It does not allow for individual differences in mast periodicity.

Third, climate variables that affect fecundity operate both through interannual anomalies over time and as [a] geographic variation. The masting literature deals almost exclusively with the former, but our application must identify the latter: the potentially smooth variation of climate effects across regions must be extracted from the many individual time series, each dominated by local "noise."

Finally, model fitting is controlled by the size classes that dominate a given site and thus is insensitive to size classes that are poorly represented. Large trees are relatively rare in eastern forests, making it hard to identify potential declines in large, old individuals[41,53]. Conversely, the shade-intolerant species that dominate second-growth forests often lack the smaller size classes needed to estimate maturation and early stages where fecundity may be increasing rapidly.

Several of the foregoing challenges are resolved in the MASTIF model by introducing latent states for individual maturation status and tree-year seed production. The dependent data types (maturation status, CCs, STs) become conditionally independent in the hierarchical MASTIF model (e.g., ref. [66]). The serial dependence is handled as a conditional hidden Markov process for maturation that combines with CCs and STs by way of stochastic (latent) conditional fecundity. Maturation status and conditional fecundity must be estimated jointly, that is, not with separate models. The latent maturation/fecundity treatment avoids imposing a specific AR($p$) structure. In the MASTIF model there is a posterior covariance in maturation/fecundity across all tree-year estimates that need not adhere to any specific assumption[20]. The dependence across individuals and years is automatic and available from the posterior distribution.

Separating the spatial from temporal components of climate effects is possible here, not only because the entire network is analyzed together but also because

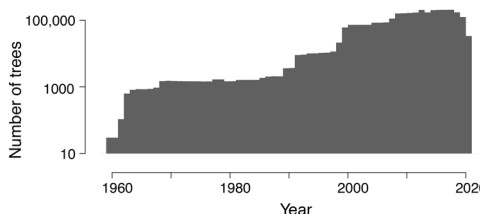

**Fig. 6 Distribution of observation trees by year in the North American region of the MASTIF network.** Sites are listed by ecoregion in the Supplementary Data 2.

predictors in the model include both climate norms for the individual sites and interannual anomalies across sites[35,52]. TA depends on both of these components.

Extracting the trends from volatile data further benefits from random individual effects for each tree and the combination of climate anomalies and year effects over time. A substantial literature focuses on specific combinations of climate variables that best explain year-to-year fecundity variation, including combinations of temperature, moisture, and water balance during specific seasons over current and previous years[19,25,41]. Results vary for each study, presumably due to the differences in sites, species, size classes, duration, data type, and modeling assumptions. For TA, the goal is to accommodate the local interannual variation to optimize identification of trends in space and time. Thus, we include a small selection of important climate anomalies (spring minimum $T$ of the current year, summer $T$ of the current and previous year, and moisture $D$ of the current and previous year). The climate anomalies considered here do not include every variable combination that could be important for all size classes of every species on every site. For this reason, we combine climate anomalies with year effects. Year effects in the model are fixed effects within an ecoregion and random between ecoregions (ecoregions are shown in Fig. 2 and listed in Supplementary Data 2). They are fixed within an ecoregion because they are not interpreted as exchangeable and drawn at random from a large population of possible years. They are random between ecoregions due to the uneven distribution of sites (Supplementary Data 1)[20].

To optimize inference on size effects, the sampling of coefficients in posterior simulation is implemented as a weighted regression. This means that the contribution of tree diameter to fecundity is inversely proportional to the abundance of that size class in the data. This approach has the effect of balancing the contributions of abundant and rare sizes. Identifying size effects further benefits from the introduction of opportunistic field sampling, which can target the large individuals that are typically absent from field study plots.

**MASTIF data network.** Data included in the analysis come from published and unpublished sources and offer one or both of the two data types, CCs and STs (Supplementary Data 1). Both data types inform tree-year fecundity; they are plotted by year in Fig. 6.

CCs in the MASTIF network are obtained by one of three methods. Most common are counts with binoculars that are recorded with an estimate of the fraction of the crop that was observed. A second CC method makes use of seeds collected per ground surface area relative to the crown area. This method is used where conspecific crowns are isolated and wind dispersal is limited. The crop fraction is the ratio of ground area for traps relative to the projected crown area. Examples include HNHR[67] and BCEF[68].

A third CC method is based on evidence for past cone production that is preserved on trees. This has been used for *Abies balsamea* at western Quebec sites[69], *Pinus ponderosa* in the Rocky Mountains[70], and for *Pinus edulis* at SW sites[27].

ST data include observations on individual trees that combine with seed counts from traps. Because individual studies can report different subcategories of seeds, and few conduct rigorous tests of viability, we had to combine them using the closest description to the concept of "viable". For example, we do not include empty conifer seeds. A dispersion model provides estimates of seeds derived from each tree. ST and CC studies are listed in Supplementary Data 1. The likelihoods for CCs and STs are detailed in ref. [20]. Individually and in combination, the two data types provide estimates, with full uncertainty, for fecundity across all tree-years.

Fitted species had multiple years of observations from multiple sites, which included 211,146 trees and 2,566,594 tree-years from 123 species. Sites are shown in Fig. 2 of the main text by ecoregion, they are named in Fig. 1 and summarized in Supplementary Data 1. For TA the fits were applied to 7,723,671 trees on inventory plots. Mean estimates for the genus were used for inventory trees belonging to species for which there were not confident fits in the MASTIF model, which amounted to 7.2% of inventory trees. Detailed site information is available at the website MASTIF.

**Table 1 Predictors in the model, not all of which are important for all species.**

| Predictors | Symbol | Dimensions | Definitions |
|---|---|---|---|
| Diameter | $G_{ij,t}$ | cm | |
| | $G_{ij,t}^2$ | cm² | $G$ squared |
| Shade | $S_{ij,t}$ | Ordinal | 1–5 FIA/NEON classes |
| $D$ | $D_j$ | cm-mo | $\sum_{m=1}^{8}(P_{jm,t} - \text{PET}_{jm,t})$ |
| $D$ anomaly | $D_{j,t}$ | cm-mo | Anomaly for site |
| Spring min $T$ | $T_{\text{sp},j}$ | °C | Mean minimum daily spring $T$ Feb–Mar |
| Spring min $T$ anomaly | $T_{\text{sp},j,t}$ | °C | Anomaly for site |
| Summer $T$ | $T_j$ | °C | Mean June–August |
| Summer $T$, quadratic | $T_j^2$ | °C² | Summer $T$ squared |
| $D$:$G$ | $D_j G_{ij,t}$ | cm-mo × cm | $D$:$G$ interaction |

Subscripts reference tree $i$ and site $j$ in month $m$ of year $t$. "sp" refers to spring.
Symbols are diameter $G$, temperature $T$, and moisture deficit $D$.

**Covariates.** Covariates in the model include as main effects tree diameter, tree canopy class (shading), and the climate variables in Fig. 1 of the main text and described in Table 1. A quadratic diameter term in the MASTIF model allows for changes in diameter response with size[52]. Shade classes follow the USDA Forest Inventory and Analysis (FIA)/National Ecological Observation Network (NEON) scheme that ranges from a fully exposed canopy that does not interact with canopies of other trees to fully shaded in the understory. Shading provides information on competition that has proved highly significant for fecundity in previous analyses[41,52].

To distinguish between the effects of spatial variation versus interannual variability, spring $T$ and moisture $D$ are included in the model as site means and site anomalies[35]. Spring minimum $T$ affect phenology and frost risk during flowering and early fruit initiation. Summer mean $T$ (June–August) is included both as a linear and quadratic term. Mean summer $T$ is linked to thermal energy availability during the growing season, with the quadratic term allowing for potential suppression due to extreme heat. Moisture $D$ (cumulative monthly PET-P (potential evapotranspiration[-] minus precipitation) for January–August) is included as a site mean and an annual anomaly. Moisture $D$ is important for carbon assimilation and fruit development during summer in the eastern continent and, additionally, from the preceding winter in the western continent. For species that develop over spring and summer, anomalies incorporate the current and previous year. We did not include longer lags in covariates. For species that disperse seed in spring (*Ulmus* spp. and some members of *Acer*), only the previous year was used. Temperature anomalies were included for spring, but not summer, simply to reduce the number of times that temperature variables enter the model, and these two variables tended to be correlated at many sites.

Climate covariates were derived from gridded climate products and combined with local climate monitoring where it is available. Terraclimate[71] provides monthly resolution, but it is spatially coarse. For both norms and trends, we used the period from 1990 to 2019 because global temperatures have been increasing consistently since the 1980s, and this span broadly overlaps with fecundity data (Fig. 6). CHELSA[72] data are downscaled to a 1 km grid, but it does not extend to 2019. Our three-component climate scaling used regression to project CHELSA forward using Terraclimate, followed by downscaling to 1 km with CHELSA, with further downscaling to local climate data. Even where local climate data exist, they often do not span the full duration of field studies, making the link to gridded climate data important. Local climate data were especially important for mountainous sites in the Appalachians, Rockies, Sierra Nevada, and Cascades.

Of the full list of variables, a subset was retained, depending on species (some have narrow geographic ranges) and deviance information criteria of the fitted model (Supplementary Data 2). Year effects in the model allow for the interannual variation that is not absorbed by anomalies[20].

**Model fitting and TA.** As mentioned above, model fitting applied the hierarchical Bayes model of ref. [20] to the combination of time series and opportunistic observations summarized in Fig. 1. Posterior simulation was completed with Markov chain Monte Carlo based on direct sampling, Metropolis, and Hamiltonian Markov chain. Model fitting used 211,146 trees and 2,566,594 tree-years from 123 species (Supplementary Data 2). Only species with multiple observation years were included.

The climate variable referenced as $C$ in Eq. (1) of the main text is, in fact, a vector of climate variables described in the previous section, spring minimum $T$, summer mean $T$, and moisture $D$ (Table 1). The anomalies and year effects in the

fitted model contribute to the trends not explained by biogeographic variation as $\gamma$ in Eq. (1). For main effects in the model, the partial derivatives are fitted coefficients, an example being the response to spring minimum temperature $\partial f/\partial T_{\text{sp}} = \beta_{T_{\text{sp}}}$. For predictors involved in interactions, the partial derivatives are combinations of fitted coefficients and variables. For example, the response to moisture $D$, which interacts with tree size, is $\partial[F]f/\partial D = \beta_D + \beta_{GD}G$. The response to diameter $G$, which is quadratic and interacts with $D$, is $\partial f/\partial G = \beta_G + 2\beta_{G^2}G + \beta_{GD}D$.

Trend decomposition applied the fitted model to every tree in forest inventories from the USDA FIA program, the Canada's National Forest Inventory, the NEON, and our MASTIF collaboration. Each tree in these inventories has a species and diameter. For trees that lack a canopy class, regression was used to predict it from distances and tree diameters based on inventories that include both location and canopy class, including NEON, FIA, and the MASTIF network. Although inventories differ in the minimum diameter they record, few trees are reproductive at diameters below the lower diameter limits in these surveys, so the effect on fecundity estimates is negligible. For the indirect effects of climate coming through tree growth rates, the same covariates were fitted to growth as previously defined for fecundity, using the change in diameter observed over multiple inventories. A Tobit model was used to accommodate the fact that a second measurement can be smaller than an earlier measurement. The Tobit thus treats negative growth as censored at zero. TA to inventory plots used 7,717,677 trees. Because not all species in the inventory data are included in the MASTIF network, mean fecundity parameters for the genus were used for unfitted species. Species fitted in the MASTIF network accounted for >90% of trees in inventory plots (Supplementary Data 2).

From the predictive distributions for every tree in the inventory data, we evaluated predictive mean trends aggregated to species and plot in Fig. 2b. We extracted specific terms that comprise the components in Fig. 4 and aggregated them too to the plot averages.

**General form for TA.** Equation 1 simplifies the model to highlight direct and indirect effects. Again, climate variables and tree size represent only a subset of the predictors in the model that are collected in a design vector $\mathbf{x}_t = [x_{1,t}, \ldots, x_{Q,t}]'$, where the $q = 1, \ldots, Q$ predictors include shading from local competition, individual size, and climate and habitat variables (Table 1). On the proportionate scale, Eq. (1) can be written in terms of all predictors, including main effects and interactions, as

$$\frac{df}{dt} = \sum_{q=1}^{Q} \left( \frac{\partial f}{\partial x_q} + \sum_{q' \in I_q} \frac{\partial f}{\partial(x_q x_{q'})} x_{q'} \right) \frac{dx_q}{dt} + \gamma \tag{2}$$

where $I_q$ are variables that interact with $x_q$. In this application, interactions include tree diameter with moisture deficit and diameter squared. Each term in the summation consists of a main effect of $x_q$ and interactions that are multiplied by the rate of change in variable $x_q$. For the simple case of only two predictors, Eq. (2) is recognizable as Eq. (1) of the main text, where $x_1, x_2$ have been substituted for variables $G$ and $C$ (Table 1). In our application, predictors include additional climate and shading (Table 1).

Recognizing that environmental variables affect not only fecundity but also growth rate, we extract the size effect, that is, the $x_q$ that is $G$, and incorporate these indirect effects (through growth) by expanding $g = dG/dt$ in Eq. (1) of the main text as

$$g = \sum_{q=1}^{Q} \left( \frac{\partial g}{\partial x_q} + \sum_{q' \in I_q} \frac{\partial g}{\partial(x_q x_{q'})} x_{q'} \right) x_q + \nu \tag{3}$$

where $\nu$ is the component of growth that is not accommodated by other terms. This expression allows us to evaluate the full effect of climate variables, including those coming indirectly through growth.

**Connecting fitted coefficients in MASTIF to TA.** This section connects the continuous, deterministic Eq. (1) to the MASTIF model[20] with the interpretation of responses, direct effects, and full effects of Fig. 5. To summarize key elements of the fitted model[20], consider a tree $i$ at site $j$ that grows to reproductive maturity and then produces seed depending on its size, local competitive environment, and climate. We wish to estimate the effects of its changing environment and condition on fecundity using a model that includes spatial variation in predictors that are tracked longitudinally over years $t$. Fecundity changes through maturation probability $\rho_{ij}(t)$, which increases as trees increase in size, and through conditional fecundity $\psi_{ij}(t)$, the annual seed production of a mature tree. Let $z_{ij}(t) = 1$ be the event that a randomly selected tree $i$ is mature in year $t$. Then, $\rho_{ij}(t)$ is the corresponding probability that the tree is mature, $E[z_{ij}(t)] = \rho_{ij}(t)$ ($\rho$ is not to be confused with the probability that a tree that is now immature will make the transition to the mature state in an interval $dt = 1$. That is a different quantity detailed in the Supplement to ref. [41]). Fecundity has expected value $F_{ij}(t) = \rho_{ij}(t)\psi_{ij}(t)$. On a proportionate (log) scale,

$$f_{ij}(t) = \log F_{ij}(t) = \log \rho_{ij}(t) + \log \psi_{it}(t) \tag{4}$$

The corresponding rate equation is

$$\frac{df}{dt} = \frac{d\log\rho}{dt} + \frac{d\log\psi}{dt} \tag{5}$$

The discretized and stochasticized version of Eq. (1) is

$$\frac{dF_{ij}}{dt} = \frac{F_{ij,t+dt} - F_{ij,t}}{dt} + \epsilon_{ij,t}$$
$$= \Delta F_{ij,t} + \epsilon_{ij,t} \tag{6}$$

where $dt = 1$ and $\epsilon_{ij,t}$ is the integration error. When applied to a dynamic process model, this term further absorbs process error (see above), which is critical here to allow for conditional independence where observations are serially dependent. In simplest terms, $\epsilon$ is model miss-specification that allows for dependence in data.

The MASTIF model that provides estimates for TA is detailed in ref. [20]. Elements of central interest for TA are

$$F_{ij,t} = z_{ij,t}\psi_{ij,t}$$
$$\left[z_{ij,t} = 1\right] \sim Bernoulli\left(\rho_{ij,t}\right)$$
$$\rho_{ij,t} = \Phi(\boldsymbol{\mu}_{ij,t})$$
$$\log\psi_{ij,t} = \mathbf{x}'_{ij,t}\boldsymbol{\beta} + h_t(T) + \epsilon_{ij,t}$$

where $\boldsymbol{\mu}_{ij,t} = \alpha_0 + \alpha_G G_{ij,t}$ describes the increase in maturation probability with size, $\Phi(\cdot)$ is the standard normal distribution function (a probit), $\epsilon_{ij,t} \sim N(0, \sigma^2)$, and $h_t(T)$ can include year effects, $h(T) = \kappa_t$, or lagged effects, $h(T) = \sum_{k=1}^{p} \kappa_k \psi_{it,t-k}$, that contribute to $\gamma$ in Eq. (1) of the main text. If year effects are used, then $\gamma$ includes the trend in year effects. (The generative version of this model writes individual states at $t$ conditional on $t - 1$ and is given in the Supplement to ref. [20].) If an AR($p$) model is used, then $\gamma = \kappa_1$ (provided data are not detrended). Random individual effects in the fitted model are marginalized for prediction of trees that were not fitted, meaning that $\sigma^2$ is the sum of model residual and random-effects variance. Again, the length-$Q$ design vector $\mathbf{x}_{ij,t}$ includes individual attributes (e.g., diameter $G_{ij,t}$), local competitive environment, and climate (Table 1). There is a corresponding coefficient vector $\boldsymbol{\beta}$.

Moving to a difference equation (rate of change) for conditional log fecundity,

$$\Delta f_{ij,t} = \Delta\log\rho_{ij,t} + \Delta\log\psi_{ij,t}$$

where

$$\Delta\log\psi_{ij,t} = \log\psi_{ij,t+1} - \log\psi_{ij,t}$$
$$= \Delta\mathbf{x}'_{ij,t}\boldsymbol{\beta} + \gamma_{ij,t} + \nu_{ij,t}$$
$$\Delta\mathbf{x}_{ij,t} = \mathbf{x}_{i,t} - \mathbf{x}_{ij,t-1}$$
$$\nu_{ij,t} \sim N(0, 2\sigma^2)$$

The variance in the last line is the variance of the difference $\Delta\epsilon_{ij,t}$.

Elements of basic theory in Eq. (1) of the main text are linked to data through the modeling framework as

$$\begin{aligned}\Delta f_{ij,t} = &+ \beta_{T_{\text{sp}}}\Delta T_{sp,j} \\ &+ \left(\beta_T + 2\beta_{T^2}T_j\right)\Delta T_j \\ &+ \left(\beta_D + \beta_{GD}G_{ij,t}\right)\Delta D_j \\ &+ \left(\alpha_G \frac{\phi(\boldsymbol{\mu}_{ij,t})}{\Phi(\boldsymbol{\mu}_{ij,t})} + \beta_G + 2\beta_{G^2}G_{ij,t} + \beta_{GD}D_j\right)\Delta G_{ij} \\ &+ \gamma_{ij,t} + \nu_{ij,t}\end{aligned} \tag{7}$$

where $\phi(\cdot)$ is the standard normal density function that comes from the rate of progress toward maturation. Again, the anomalies do not appear in this expression for trends because trends in the anomalies and year effects enter through $\gamma$.

The first four lines in Eq. (7) are, respectively, the effects of trends in spring minimum temperatures $\Delta T_{sp,j}$, summer mean temperature $\Delta T_j$, moisture deficit $\Delta D_j$, and size $\Delta G_{ij}$, where the latter comes from growth on inventory plots. The contribution of maturation to change in fecundity is the first term in the fourth line, $\alpha_G \phi(\boldsymbol{\mu}_{ij,t})/\Phi(\boldsymbol{\mu}_{ij,t})$. A map of this term in Fig. 7b shows the strong contribution to fecundity in the East due to the young (Fig. 7a) and/or small (Fig. 4b) trees there. The sum of these terms dominates the patterns in Fig. 3c.

All terms in Eq. (7) have units of mean change in proportionate fecundity, and these are mapped in figures of the main text. We focus on proportionate fecundity because it reflects the full effect of climate as opposed to total fecundity, which would often be dominated by one or a few trees of a single species. However, from proportionate fecundity we can obtain change in fecundity as $\Delta F_{ij,t} = F_{ij,t} \times \Delta f_{ij}$. Stand-level effects on fecundity change at site $j$ can be obtained from individual change as

$$\Delta F_j = \sum_{i=1}^{n_j} \Delta F_{ij} = \sum_{i=1}^{n_j} F_{ij}\Delta f_{ij,t}$$

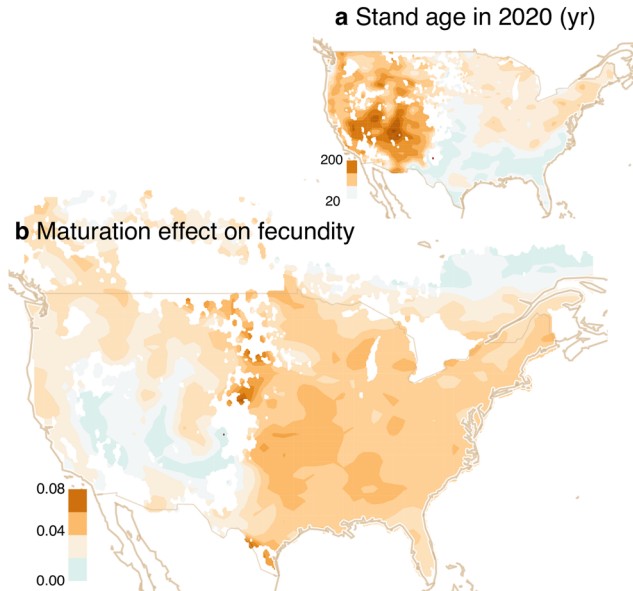

**a** Stand age in 2020 (yr)

**b** Maturation effect on fecundity

**Fig. 7 Size and maturation effects on fecundity. a** Stand age variable in FIA data and **b** positive effect of maturation for increasing fecundity in the eastern continent. In the West, maturation does not contribute to rising fecundity because large trees are predominantly [mature] larger.

Again, maps in Fig. 5 show mean proportionate effects for all trees on an inventory plot.

**Reporting summary**. Further information on research design is available in the Nature Research Reporting Summary linked to this article.

## Data availability
Data from the study are available at the Duke Data Repository (Dataset) [https://doi.org/10.7924/r4348ph5t].

## Code availability
Code in R and C++ is available on CRAN as mastif version 1.0.1 [https://cran.r-project.org/web/packages/mastif/index.html], with additional background here.

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

## Acknowledgements

For access to sites and logistical support, we thank the National Ecological Observatory Network (NEON). For comments on the manuscript, we thank Valentin Journe and Becky Tang. The project was funded for three decades by the National Science Foundation (most recently, DEB-1754443), and by the Belmont Forum (1854976), NASA (AIST16-0052, AIST18-0063), and the Programme d'Investissement d'Avenir under project FORBIC (18-MPGA-0004). Any use of trade, firm, or product names is for descriptive purposes only and does not imply endorsement by the US Government.

## Author contributions

All authors collected or provided data and revised the paper. J.S.C. developed the model, compiled the data, wrote the software, and wrote the paper.

## Competing interests

The authors declare no competing interests.

## Additional information

[1]Nicholas School of the Environment, Duke University, Durham, NC, USA. [2]INRAE, LESSEM, University Grenoble Alpes, Saint-Martin-d'Heres, France. [3]Department of Geography, University of Colorado Boulder, Boulder, CO, USA. [4]School of Natural Sciences, University of California, Merced, Merced, CA, USA. [5]Forest Research Institute, University of Quebec in Abitibi-Temiscamingue, Rouyn-Noranda, QC, Canada. [6]Department of Systematic Zoology, Faculty of Biology, Adam Mickiewicz University, Poznan, Poland. [7]USDA Forest Service, Southern Research Station, Monticello, AR, USA. [8]USDA Forest Service Southern Research Station, Auburn, AL, USA. [9]Natural Resources, Cornell University, Ithaca, NY, USA. [10]Institute for the Environment, University of North Carolina at Chapel Hill, Chapel Hill, NC, USA. [11]Greater Yellowstone Network, National Park Service, Bozeman, MT, USA. [12]USGS Western Ecological Research Center, Three Rivers, CA, USA. [13]Earth and Environment, Boston University, Boston, MA, USA. [14]Finnish Meteorological Institute, Helsinki, Finland. [15]Forest Resources, University of Washington, Seattle, WA, USA. [16]Department of Biological Science, Northern Arizona University, Flagstaff, AZ, USA. [17]University of California, Santa Cruz, Santa Cruz, CA, USA. [18]USDA Forest Service, Bent Creek Experimental Forest, Asheville, NC, USA. [19]USDA Forest Service Southern Research Station, Eastern Forest Environmental Threat Assessment Center, Research Triangle Park, NC, USA. [20]Department of Biology, University of Washington, Seattle, WA, USA. [21]School for Environment and Sustainability, University of Michigan, Ann Arbor, MI, USA. [22]Department of Biology, University of Saskatchewan, Saskatoon, SK, Canada. [23]Health and Environmental Sciences Department, Xian Jiaotong-Liverpool University, Suzhou, China. [24]Hastings Reservation, University of California Berkeley, Carmel Valley, CA, USA. [25]Department of Biological Sciences, DePaul University, Chicago, IL, USA. [26]Department of Wildland Resources, Utah State University Ecology Center, Logan, UT, USA. [27]Department of Biology, University of New Mexico, Albuquerque, NM, USA. [28]Pacific Forestry Centre, Victoria, BC, Canada. [29]Université du Québec en Abitibi-Témiscamingue, Rouyn-Noranda, Quebec, Canada. [30]Department of Biology, Colby College, Waterville, ME, USA. [31]Department of Biology, Washington University in St. Louis, St. Louis, MO, USA. [32]University of New Mexico, Albuquerque, NM, USA. [33]Department for the Ecology of Animal Societies, Max Planck Institute of Animal Behavior, Konstanz, Germany. [34]Valles Caldera National Preserve, National Park Service, Jemez Springs, NM, USA. [35]Fort Collins Science Center, Fort Collins, CO, USA. [36]Department of Natural Sciences, Mars Hill University, Mars Hill, NC, USA. [37]Department of Forest and Rangeland Stewardship, Colorado State University, Fort Collins, CO, USA. [38]Ecology and Evolutionary Biology, University of Toronto, Toronto, ON, Canada. [39]Department of Biology, Wilkes University, Wilkes-Barre, PA, USA. [40]Geography Department and Russian and East European Institute, Bloomington, IN, USA. ✉email: jimclark@duke.edu

