## [Peer Review File · Nature Communications]

Reviewers' Comments:

Reviewer #1:

Remarks to the Author:

The results of this analysis are exceptionally novel and important, and the dataset and analysis are robust. Demonstrating that climate change effects on forest fecundity will be driven mainly by indirect effects on tree size (and the implication for other demographic processes too) is a real game-changer. Demonstrating that differences in size structure of forests in the east and west of USA result will result in divergent near-term trends in fecundity as a result of climate change is also a key result. Overall, this is fundamentally important paper that will change the way that we consider climate change effects on forest fecundity, but also more widely on the response of forests to climate change. This has the potential to be a superb paper, and to be very well suited to the journal.

The guidelines to reviewers indicate that published paper should meet several criteria:

- The data is technically sound
- The paper provides strong evidence for its conclusions
- The results are novel
- The manuscript is important to scientists in the specific field

The manuscript clearly meets, and indeed exceeds, all of these criteria. The data is exceptional, and the methods used to analyze it are robust and appropriate (although also quite hard to follow for a non-specialist in the methods). The results are highly novel – identifying and then demonstrating the indirect effects of climate change on tree size and its subsequent effect on fecundity is a major step forward for the field, and has widespread implications for the field. The paper can be a real game-changer.

Having said this, I have found this a very difficult manuscript to read and review. The key results and their implications are buried under dense text, such that the paper is very hard to follow. While I realize that the journal is focused on important research for specialized research fields, I think the authors need to make it easier for the reader to follow the argument – this will maximize the impact of their crucial new insights. I make some specific points below, but overall I suggest that the paper needs a full and careful revision of the text with the (specialist) reader in mind.

* some examples of the challenges currently presented to the reader *

With careful study, Figure 3 indicates very interesting and novel results. For example panel (b) indicates that the direct effect of moisture deficit is negative across most of North America – this is the type of result that might originate from a “simple” longitudinal analysis of fecundity datasets. However, the strong indirect effect of higher tree growth rates in eastern NA overcomes this, such that the full effect of change in moisture deficit on fecundity is actually positive in eastern NA. However, during my initial read through the manuscript, I missed this point entirely, while I tried to follow the difficult-to-follow introduction of background, methods and results. This is just one example, but it illustrates a key weakness in the manuscript - it forces the reader to work very hard to appreciate the significance of the manuscript.

In Figure 3 the reader is encouraged to compare the magnitude of the direct (b) and indirect (c) effects. Careful study reveals the remarkable difference (and dominance of the indirect effects), but this would be much easier to recognize if the color scales were consistent across the plots (at least in the rows, recognizing this may not be possible or appropriate for (a)).

The manuscript is heavily focused on the methodology used, and highlighting some of the issues with previous approaches (e.g. meta-analysis). This is fine, I don't disagree, but this discussion is given undue prominence considering the very exciting and important scientific results presented. The paper struggles to grapple with not having a Methodology section up front, and instead tries to squeeze this in to the other sections – it all ends up rather untidy and difficult to follow.

MASTIF is very confusing. Initially it is introduced as a network, then later as a project. At times it appears to be a dataset of seed production data (longitudinal studies plus the data from iNaturalist), but sometimes it appears to be the wider project incorporating the data and the modelling framework. This needs clarification throughout the manuscript, I don't understand what MASTIF is.

* Some scientific points *

Seed production is not the same as fecundity, although I am not arguing that the exceptional datasets assembled by the authors are not suitable for the proposed analysis. Instead, I suggest the authors should consider some acknowledgement that viable seed production might be a better measure – although I realize that such data is not available for the present analysis (this is absolutely not a criticism of the data or analysis, just a comment on its interpretation). Seed quantity varies between years. This can be as a function of masting in species with highly variable seed production, but also in response to variation in the environmental variables analyzed in the paper – e.g. climate change may not only affect the number of seeds, as analyzed here, but also the proportion pollinated, the proportion that are viable, and the seed quality (i.e. germination success of apparently viable seeds).

The paper is focused on climate change – and this is understandable – but changes in CO₂ and nutrient availability will also influence fecundity directly and indirect through the outlined mechanisms. It might be beyond the scope of this analysis to include these additional effects, but they should be acknowledged. In terms of tree growth rates, these factors could have larger effects than climate?

The insight that climate-driven changes in tree size will influence seed production is very important, and a key strength of the manuscript. Have the authors considered the potential for changes in within-tree allocation to growth, reproduction etc.? This happens as a function of size, as demonstrated in the manuscript, but could this also reflect a change in tree nutritional status for example?

Time-periods of analysis are very vague in the manuscript. I recognize that these will vary between sites etc., but climate trends are highly dynamic and will be very sensitive to the selected time interval. Figure 1 indicates that climate trends have been calculated since 1985, but until what year?

The final section provides some thoughts on the implications for managing resilient forests. The results imply that to manage forests a strategy that maximizes fecundity might be preferable - or at least ensures that fecundity is maintained. However, this appears to conflict with the significance of large and old trees, which have passed peak fecundity but have high value in other respects (e.g. carbon sequestration, biodiversity, cultural value). Additionally, is there an implication that forests in West are "overmature", to the detriment of their ability to cope with climate change through biogeographic shifts? The paper hints that perhaps these forests should be managed to restore fecundity (and the capacity of fecundity to keep pace with climate change)?

Reviewer #2

Review of 'Continent-wide tree fecundity driven by indirect climate effects' NCOMMS-20-31693-T

This study attempts to predict the impact of changing climate in North America on tree fecundity, both as a direct effect and also through indirect effects on tree growth. The latter is derived from some understanding of the relationship between tree size and fecundity. Animals lose fecundity as they get older and this is also appreciated for trees, although data are scarce. As recognised by the authors these processes are complicated by the phenomenon of masting which is an occasional massive increase in fecundity exhibited by many tree species. The authors attempt to overcome some of these potential constraints by 'introducing latent states for individual maturation status and tree-year seed production...' I must confess I cannot penetrate the methodology but fail to imagine how model sophistication would be able to cope with limitations in data relating to an adequate understanding of seed production through time in relation to tree size. The authors make it clear that data is limited for key parameters including masting and the relationship between tree size (age) and fecundity. I draw attention to the study of Fung and Waples (2017) which also highlights the deficiency of the latter data. My failure to comprehend the calculus and modelling required to generate the conclusions of this study are a major short coming of my review.

However, I can make another comment. The first sentence states 'The composition and structure of twenty-first century forests will depend on the seed production needed for tree populations to keep pace with climate change.' I am not so sure. My understanding of the ecological literature is that seed production is not a major constraint on the composition and structure of forests. A lot happens in forest dynamics after seed production. A fairly recent study from Australia (Fensham et al. 2015) indicates that drought-induced tree mortality was selective towards the dominant species suggesting a substantial change in species composition. Similar suggestions have been made in Europe and North America (Allen and Breshears 1998). However, the Australian study also demonstrated that the dominant trees have abundant regenerative capacity (in the form of small trees). Thus there does not seem to be a bottle-neck in the abundance of small trees to result in a lasting effect on composition. For seed production to be a critical determinant of structure and composition it needs to be demonstrated that the availability of seed has the potential to be a bottle-neck. There is not compelling demonstration of this bottle-neck in this study. There is an argument that the presence of masting in tree species is an evolutionary response to ensure that such a bottle-neck is not a constraint on the success of a species.

Despite my circumspection the authors represent a substantial body of data through a model that I cannot comprehend and present a well-written argument for another plank of climate change concern, namely the simultaneous increasing and decreasing of tree fecundity. Thus their method and conclusions may well be robust and worthy of publication in a high profile journal such as Nature Communications.

Allen CD, Breshears DD (1998) Drought-induced shift of a forest-woodland ecotone: Rapid landscape response to climate variation. *Proc Natl Acad Sci USA* 9:14839–14842

Fensham RJ, Fraser J, Macdermott HJ, Firn J (2015) Dominant tree species are at risk from exaggerated drought under climate change. *Glob Chang Biol* 21:3777–3785.

<https://doi.org/10.1111/gcb.12981>

Fung HC, Waples RS (2017) Performance of IUCN proxies for generation length. *Conserv Biol* 31:883–893. <https://doi.org/10.1111/cobi.12901>

REVIEWER COMMENTS

Reviewer #1 (Remarks to the Author):

The results of this analysis are exceptionally novel and important, and the dataset and analysis are robust. Demonstrating that climate change effects on forest fecundity will be driven mainly by indirect effects on tree size (and the implication for other demographic processes too) is a real game-changer. Demonstrating that differences in size structure of forests in the east and west of USA result will result in divergent near-term trends in fecundity as a result of climate change is also a key result. Overall, this is fundamentally important paper that will change the way that we consider climate change effects on forest fecundity, but also more widely on the response of forests to climate change. This has the potential to be a superb paper, and to be very well suited to the journal.

The guidelines to reviewers indicate that published paper should meet several criteria:

November 8, 2019

- The data is technically sound
- The paper provides strong evidence for its conclusions
- The results are novel
- The manuscript is important to scientists in the specific field

The manuscript clearly meets, and indeed exceeds, all of these criteria. The data is exceptional, and the methods used to analyze it are robust and appropriate (although also quite hard to follow for a non-specialist in the methods). The results are highly novel – identifying and then demonstrating the indirect effects of climate change on tree size and its subsequent effect on fecundity is a major step forward for the field, and has widespread implications for the field. The paper can be a real game-changer.

Having said this, I have found this a very difficult manuscript to read and review. The key results and their implications are buried under dense text, such that the paper is very hard to follow. While I realize that the journal is focused on important research for specialized research fields, I think the authors need to make it easier for the reader to follow the argument – this will maximize the impact of their crucial new insights. I make some specific points below, but overall I suggest that the paper needs a full and careful revision of the text with the (specialist) reader in mind.

* some examples of the challenges currently presented to the reader *

With careful study, Figure 3 indicates very interesting and novel results. For example panel (b) indicates that the direct effect of moisture deficit is negative across most of North America – this is the type of result that might originate from a “simple” longitudinal analysis of fecundity datasets. However, the strong indirect effect of higher tree growth rates in eastern NA overcomes this, such that the full effect of change in moisture deficit on fecundity is actually positive in eastern NA. However, during my initial read through the manuscript, I missed this point entirely, while I tried to follow the difficult-to-follow introduction of background, methods and results. This is just one example, but it illustrates a key weakness in the manuscript - it forces the reader to work very hard to appreciate the significance of the manuscript.

We agree that this is a difficult concept, and we have revised carefully to help clarify. We have expanded the section on the interpretation of Fig. 3, which now begins on line 117. Beginning on line 125 we have emphasized that an environmental variable has a positive effect whenever the response and the direction of change **have the same sign**. We believe that the additional explanation in this section brings clarity.

In Figure 3 the reader is encouraged to compare the magnitude of the direct (b) and indirect (c) effects. Careful study reveals the remarkable difference (and dominance of the indirect effects), but this would be much easier to recognize if the color scales were consistent across the plots (at least in the rows, recognizing this may not be possible or appropriate for (a).

We wondered about this too but found that we could not apply the same scale to different maps, because only those responses and effects that vary a lot would show any spatial effects. We have incorporated additional text beginning on line 118 to emphasize the scale differences between maps.

The manuscript is heavily focused on the methodology used, and highlighting some of the issues with previous approaches (e.g. meta-analysis). This is fine, I don't disagree, but this discussion is given undue prominence considering the very exciting and important scientific results presented. The paper struggles to grapple with not having a Methodology section up front, and instead tries to squeeze this in to the other sections – it all ends up rather untidy and difficult to follow.

November 8, 2019

We agree and have added text to frame its relevance in both the Introduction and the Discussion. The Discussion has been expanded to emphasize how these results contribute to the alternative methods now in the literature, a paragraph that begins on line 163. The paragraph beginning on line 175 focuses on how TA expands our ability to interpret the causes for responses. To the Introduction we added a section beginning on line 20 on recruitment limitation, why fecundity is basically absent from most models, and why this is important. We revisit this issue in the new paragraph beginning at line 185. Although we avoided eliminating critical technical details, we feel that these additions shift the focus to the scientific results.

MASTIF is very confusing. Initially it is introduced as a network, then later as a project. At times it appears to be a dataset of seed production data (longitudinal studies plus the data from iNaturalist), but sometimes it appears to be the wider project incorporating the data and the modelling framework. This needs clarification throughout the manuscript, I don't understand what MASTIF is.

We agree and have edited to qualify the term with "network", "project", or "model". We have done this in the main text and in the Supplement.

* Some scientific points *

Seed production is not the same as fecundity, although I am not arguing that the exceptional datasets assembled by the authors are not suitable for the proposed analysis. Instead, I suggest the authors should consider some acknowledgement that viable seed production might be a better measure – although I realize that such data is not available for the present analysis (this is absolutely not a criticism of the data or analysis, just a comment on its interpretation). Seed quantity varies between years. This can be as a function of masting in species with highly variable seed production, but also in response to variation in the environmental variables analyzed in the paper – e.g. climate change may not only affect the number of seeds, as analyzed here, but also the proportion pollinated, the proportion that are viable, and the seed quality (i.e. germination success of apparently viable seeds).

This is an important point, and we have applied the best decisions we could to standardize across data sets collected in many different ways. Each study can apply slightly different criteria for seeds. To combine them we had to select categories most consistent with the notion of 'viable'. For some this was as simple as whether or not a seed was filled, but there was variability in the definitions. We have added this explanation to Section S1.2.

The paper is focused on climate change – and this is understandable – but changes in CO₂ and nutrient availability will also influence fecundity directly and indirect through the outlined mechanisms. It might be beyond the scope of this analysis to include these additional effects, but they should be acknowledged. In terms of tree growth rates, these factors could have larger effects than climate?

This is an important point. We have added this to the Discussion at line 181.

The insight that climate-driven changes in tree size will influence seed production is very important, and a key strength of the manuscript. Have the authors considered the potential for changes in within-tree allocation to growth, reproduction etc.? This happens as a function of size, as demonstrated in the manuscript, but could this also reflect a change in tree nutritional status for example?

We agree that this is an important point. Indeed, these within-tree allocation relationships have dominated the masting literature, clearly showing that such tradeoffs can occur and sometimes be

November 8, 2019

prominent. We had previously cited this literature where we pointed out the distinction between velocity trends and interannual fluctuations. We have expanded on this point in the revised version, where this section starts on line 104. Here we also incorporated some material that was previously in the Supplement demonstrating how TA builds from the concept of climate velocity, which, again, uses information on the variability in climate responses and climate trends.

Time-periods of analysis are very vague in the manuscript. I recognize that these will vary between sites etc., but climate trends are highly dynamic and will be very sensitive to the selected time interval. Figure 1 indicates that climate trends have been calculated since 1985, but until what year?

To clarify the time interval included in fecundity data, we have added a new figure S1.2. For the Trend Attribution we used the time interval from 1990, when global temperatures began to increase at roughly the current rate. This also aligns well with data in the network. We have added this information to the Supplement.

The final section provides some thoughts on the implications for managing resilient forests. The results imply that to manage forests a strategy that maximizes fecundity might be preferable - or at least ensures that fecundity is maintained. However, this appears to conflict with the significance of large and old trees, which have passed peak fecundity but have high value in other respects (e.g. carbon sequestration, biodiversity, cultural value). Additionally, is there an implication that forests in West are “overmature”, to the detriment of their ability to cope with climate change through biogeographic shifts? The paper hints that perhaps these forests should be managed to restore fecundity (and the capacity of fecundity to keep pace with climate change)?

We did not intent to promote management actions that would threaten old-growth, nor would our results support that. We have added a paragraph to the Discussion (line 200) clarifying that management that promotes one is likely to also promote the other.

Reviewer #2 (Remarks to the Author):

Review of ‘Continent-wide tree fecundity driven by indirect climate effects’ NCOMMS-20-31693-T

This study attempts to predict the impact of changing climate in North America on tree fecundity, both as a direct effect and also through indirect effects on tree growth. The latter is derived from some understanding of the relationship between tree size and fecundity. Animals lose fecundity as they get older and this is also appreciated for trees, although data are scarce. As recognised by the authors these processes are complicated by the phenomenon of masting which is an occasional massive increase in fecundity exhibited by many tree species. The authors attempt to overcome some of these potential constraints by ‘introducing latent states for individual maturation status and tree-year seed production...’ I must confess I cannot penetrate the methodology but fail to imagine how model sophistication would be able to cope with limitations in data relating to an adequate understanding of seed production through time in relation to tree size. The authors make it clear that data is limited for key parameters including masting and the relationship between tree size (age) and fecundity. I draw attention to the study of Fung and Waples (2017) which also highlights the deficiency of the latter data. My failure to comprehend the calculus and modelling required to generate the conclusions of this study are a major short coming of my review.

We greatly appreciate the concerns raised here and hope we have increased the accessibility of the analysis. We have carefully revised the entire ms with special attention to clarity on methods. We

November 8, 2019

hope that by confining technical material to the Supplement that a non-technical audience can choose to overlook details and focus instead on the key patterns displayed by maps.

The one technical aspect that appears in the main text is the pde (eqn 1). We understand that this will not be transparent for all ecologists, that fact alone contributing to the lack of interactions in the current literature. A solid majority of our 64 authors felt that eqn 1 needed to be included in the main text, and we make additional efforts to clearly explain its message throughout the revised text.

However, I can make another comment. The first sentence states ‘The composition and structure of twenty-first century forests will depend on the seed production needed for tree populations to keep pace with climate change.’ I am not so sure. My understanding of the ecological literature is that seed production is not a major constraint on the composition and structure of forests. A lot happens in forest dynamics after seed production. A fairly recent study from Australia (Fensham et al. 2015) indicates that drought-induced tree mortality was selective towards the dominant species suggesting a substantial change in species composition. Similar suggestions have been made in Europe and North America (Allen and Breshears 1998). However, the Australian study also demonstrated that the dominant trees have abundant regenerative capacity (in the form of small trees). Thus there does not seem to be a bottle-neck in the abundance of small trees to result in a lasting effect on composition. For seed production to be a critical determinant of structure and composition it needs to be demonstrated that the availability of seed has the potential to be a bottle-neck. There is not compelling demonstration of this bottle-neck in this study. There is an argument that the presence of masting in tree species is an evolutionary response to ensure that such a bottle-neck is not a constraint on the success of a species.

As this review points out, our study does not address whether or not forests are seed-limited. We do agree with this reviewer on the importance of forest diebacks, as was highlighted in the third sentence of our paper. We hope we have not misunderstood this reviewer, because the Fensham study she/he mentions would typically be interpreted as extreme recruitment limitation: of 21 species presented in Fensham’s table 1, only one third report seedling-to-adult ratios as high as 2—to be clear, that’s only two seedlings to replace an adult tree decades into the future. Given the large loss rates from seedling to adult stages, even their most extreme ratio of 16 would not inspire confidence in stand replacement.

Having said this, we believe that the reviewer raises an important question that could be shared by others. To address this specifically, we have added a paragraph to the Introduction (line 20) and the Discussion (line 185) that cite representative papers on seed limitation. We go further to point out that the lack of understanding of fecundity’s role is so poor as to be essentially omitted from the stand simulators widely used to predict future forests. The built-in assumptions of those models include a continuous supply of seedlings, which insures that diversity does not collapse.

Despite my circumspection the authors represent a substantial body of data through a model that I cannot comprehend and present a well-written argument for another plank of climate change concern, namely the simultaneous increasing and decreasing of tree fecundity. Thus their method and conclusions may well be robust and worthy of publication in a high profile journal such as Nature Communications.

We greatly appreciate the thoughtful comments and believe they have improved the paper.

Allen CD, Breshears DD (1998) Drought-induced shift of a forest-woodland ecotone: Rapid landscape response to climate variation. Proc Natl Acad Sci USA 95:14839–14842

November 8, 2019

Fensham RJ, Fraser J, Macdermott HJ, Firn J (2015) Dominant tree species are at risk from exaggerated drought under climate change. *Glob Chang Biol* 21:3777–3785.

<https://doi.org/10.1111/gcb.12981>

Fung HC, Waples RS (2017) Performance of IUCN proxies for generation length. *Conserv Biol* 31:883–

893. <https://doi.org/10.1111/cobi.12901>

Reviewers' Comments:

Reviewer #2:

Remarks to the Author:

I have read through the authors response to reviews, and am impressed that the authors have carefully considered and responded appropriately. The original manuscript was very substantial and considered, and the authors have extended this approach with their response. I am particularly impressed that they have attempted to respond to the suggestions of improving the transparency of the analysis. This is an important study using an extremely impressive dataset.